# Analytical Evaluation and Experiment of the Dynamic Characteristics of Double-Thimble-Type Fiber Bragg Grating Temperature Sensors

**DOI:** 10.3390/mi12010016

**Published:** 2020-12-26

**Authors:** Chuan Luo, Han Wang, Dacheng Zhang, Zhengang Zhao, Yingna Li, Chuan Li, Ke Liang

**Affiliations:** 1Faculty of Information Engineering and Automation, Kunming University of Science and Technology, Kunming 650500, China; luochuan@kust.edu.cn (C.L.); wangh@stu.kust.edu.cn (H.W.); dacheng.zhang@kust.edu.cn (D.Z.); liyingna@kust.edu.cn (Y.L.); lichuan@kust.edu.cn (C.L.); liangke@stu.kust.edu.cn (K.L.); 2Yunnan Key Laboratory of Computer Technology Applications, Kunming University of Science and Technology, Kunming 650500, China

**Keywords:** fiber Bragg grating (FBG) temperature sensor, double-thimble, response speed, LHC method, numerical simulation, Biot characteristic number

## Abstract

A double-thimble-type fiber Bragg grating (FBG) temperature sensor that isolates the stress strain is developed, and the three materials of air, grease, and copper thimble are employed for encapsulating. To investigate the effect of different encapsulation materials on the time constant of the sensors under dynamic conditions, the transient heat conduction mathematical model is built according to the lumped heat capacity (LHC) system and thermal equilibrium theory, and the time constant is solved by an analytical solution. Then, a proportional three-dimensional sensor simulation model is established and the transient heat transfer process is numerically solved by the finite element analysis method. To verify the models, an experimental system is established to test the response speed of the three-type sensor and the experimental data are compared with the analytical and numerical solution results. The results show that the dynamic response performance depends on the encapsulation material parameters; the response speed is faster than recovery speed; and the response speed of the air packaging sensor is more than 20% faster than that of the grease packaging sensor, and more than 30% faster than that of the copper packaging sensor. The smaller the heat storage capacity and the larger the heat transfer coefficient, the faster the sensor’s response speed.

## 1. Introduction

Fiber grating is an optical waveguide that uses the ultraviolet photosensitive effect of doped fiber to form a spatial phase grating in the core. When light passes through the grating, the light satisfying the Bragg wavelength condition is reflected or transmitted. The wavelength drift of the fiber Bragg grating (FBG) reflection spectrum is directly related to temperature changes, so the FBG temperature sensor is one of the most important and direct application directions of fiber-grating-sensing technology [1]. FBG temperature sensors are used in many applications, such as strong magnetic fields, electric power plants, etc., due to the advantages of antielectromagnetic interference, wide dynamic range, strong reliability, and fast response performance [2,3]. Most commonly, the FBG temperature sensor is buried in the concrete to track the groundwater temperature field of ultra-high hydropower dams, and buried in the rock layer soil to monitor the frost heaving environment of the soil in the alpine tunnel and slope. However, fiber gratings are very fragile and extremely sensitive to temperature shift and stress shift, so, the grating must be packaged appropriately to be practical. The encapsulation structure of FBG temperature sensors must provide excellent mechanical strength to effectively isolate the interference of stress and strain in a complex environment. In addition, to achieve response to the ambient temperature rapidly, good thermal conductivity of the encapsulation structure is required [4,5,6]. In most cases however, these two properties are contradictory. Excellent strength inevitably leads to a thicker encapsulation structure, thus, affecting the response speed. In contrast, a protective structure with good thermal conductivity is required to achieve a rapid response, which reduces the strength of the sensor [7]. There is great engineering value in researching the encapsulation technology of temperature sensors and to study the relationship between the temperature sensors’ dynamic response performance and their encapsulation structure and materials.

A few researchers discussed the dynamic response speed of FBG temperature sensors. The regenerated FBG (RFBG) can be used for measurement up to 1100 ${}^{{}^{\circ}}$C, but the annealing process required to regenerate such gratings makes the optical fiber too delicate, so David Barrera et al. (2012) [8] protected the fiber grating with a ceramic tube, which in turn, is shielded with a thick metal casing, and performed a characterization of the thermo-optical response of both packaged and unpackaged RFBG sensors. The response and recovery times of packaged sensors were found to be 9 s and 22 s, respectively. In another study, aimed at the requirement for a fast-response expendable ocean temperature sensor, Zhang et al. (2014) [9] presented an FBG temperature sensor packaged with a mixture and metal tube; the sensor is less than 15 mm and the response time is 48.6 ms, which is an order of magnitude greater than that of an ordinary optical fiber temperature sensor. Chen et al. (2015) [10], designed a double-thimble-type FBG temperature sensor that is insensitive to external stress and thermal strain by taking into account the thermal expansion characteristics of the encapsulation structure; then, the sensor was tested by the water bath method, response time was 50∼60 s when the temperature jumped from 20 ${}^{{}^{\circ}}$C to 70 ${}^{{}^{\circ}}$C. Taking fire monitoring as an example, Rinaudo et al. (2016) [11] proposed a methodology to estimate the response time of sensors based on an analytical model of heat transfer between the sensor and its surroundings. The methodology is applied to estimate the response time of three different high-temperature fiber optic sensors developed, then, the method is validated by an experimental study. However, these research methods and objects are aimed at specific application environments and have not been promoted, so there is no general standard to quantify the response speed in fiber-optic sensors [12].

In this work, a double-thimble-type FBG temperature sensor isolating the stress and strain is designed to study the influence of encapsulation structure and materials on the response speed. Three materials of air, grease, and copper thimble are used to encapsulate the sensor. To analyze the dynamic response characteristics of these sensors, based on the Biot characteristic number ($B_{i}$) and the principle of the lumped heat capacity (LHC) system, the transient heat conduction solution model of the sensor is established and the dynamic response speed is solved. The numerical model of the temperature sensor is built and a finite element simulation is carried out to obtain the temperature field of the sensor changes with time. Finally, the temperature sensors with three package structures are processed and an experimental system is built to perform the static calibration and dynamic testing of sensors. Combining the package structure and experimental data, the influencing factors of the sensor’s dynamic response characteristics are analyzed.

## 2. Structure and Principle

### 2.1. Structure and Encapsulation

The spatial-phase Bragg gratings were fabricated using the phase mask method in the single-mode optical fiber core to form the narrowband filter to reflect and transmit the light signal of a specific wavelength, as shown in Figure 1. The measured variables (temperature, strain, etc.) of the environment can be measured by analyzing the filtered optical signal.

The Bragg wavelength $\lambda_{B}$ depends both on physical characteristics (the effective refractive index of the reverse coupling mode, $n_{eff}$) of the fiber and the geometrical characteristics (period of the grating, Λ) of the grating [13,14], the relation between these elements satisfies the light equation
(1)$$\lambda_{B}=2n_{eff}\Lambda .$$

It shows that the changes in $n_{eff}$ and Λ will cause the center wavelength shift of the reflection spectrum. Since photoelastic and thermoelastic tensors in optical fibers are not null, the temperature causes the thermal expansion effect, thermo-optical effect, and internal thermal stress of the fiber [15,16,17], meanwhile, the stress and strain cause the elastic deformation and elastic–optical effect of the optical fiber [18]. Therefore, both temperature variations and mechanical strain variations affect the refractive index and grating period, which ultimately affect the central wavelength of reflection spectrum. The Bragg wavelength shift $\Delta \lambda_{B}$ is given by
(2)$$\Delta \lambda_{B}=2n_{eff}\Delta \Lambda +2\Delta n_{eff}\Lambda .$$

Therefore, when designing an FBG temperature sensor for use in a complex stress environment, the influence of stress and strain on the Bragg wavelength shift must be eliminated. In this work, a double-thimble-type FBG temperature sensor is designed, the structure diagram is shown in Figure 2. The outer thimble is made of high-strength $304^{\#}$ stainless steel with an outer diameter of 4.5 mm, which has good formability and corrosion resistance, and can effectively resist the impact of mechanical forces in the construction process; therefore, it can ensure the survival rate and service life of the sensor in harsh environments. The inner thimble is a capillary drawn from red copper material, which has good thermal conductivity and flexibility, the outer and inner diameter are 1.5 mm and 0.3 mm, respectively. Both ends of the outer thimble are connected to the flange plate by threads. One end of the inner thimble is bonded to the flange plate by epoxy resin, the other is a free end, which is not affected by external forces. The space between the two thimbles is used as a filling layer for the thermally conductive material. The stainless steel thimble uniformly conducts heat and isolates the applied stress and strain, while the inner thimble isolates the apparent thermal strain of the encapsulation structure and protects the grating [4]. The optical fiber passes through the capillary and is fixed in it with epoxy resin. To prevent the thermal expansion effect of the inner thimble from affecting the wavelength drift of the grating, the length of the optical fiber ($l_{o}$) is slightly larger than the length of the capillary ($l_{c}$), so, the reserved length (*l*) of the fiber is
(3)$$l_{o}-l_{c}=l>C_{e}l_{c}\alpha \Delta T,$$
where α is the thermal expansion coefficient of the copper capillary, $l_{c}$ is the length of the copper capillary (60 mm), and $C_{e}$ is the engineering coefficient. For the copper with thermal expansion coefficient $\alpha =17\times 10^{-6}$
${}^{{}^{\circ}}$C${}^{-1}$, within the temperature range of 200 ${}^{{}^{\circ}}$C, the reserved optical fiber length *l* should be greater than 0.4 mm. The grating is located in the middle of the inner thimble and is in a relaxed state without tensile strain and compressive strain.

This structure is expected to result in two major problems in practical application: one is the sensor’s strength, the other is the response speed. Excellent strength inevitably leads to a thicker package structure. In contrast, an encapsulation structure with good thermal conductivity reduces the strength of the sensor. However, a specially packaged FBG sensor requires higher sensitivity and smaller time constant. Given the above situation, three encapsulation structures are designed. One is where air is used as the heat transfer medium between the outer thimble and the inner thimble, hereinafter referred to as an air-medium FBG sensor. According to the Grashof criterion [19], when the fluid flow is restricted in a limited interlayer space and the development of air thermal boundary layer is disturbed, the heat transfer method of air is heat conduction instead of heat convection. The other is to fill thermally conductive silicone grease in the filling layer, hereinafter referred to as the grease-medium FBG sensor. Thermal grease has good thermal conductivity properties, and the heat transfer mode is heat conduction. The third type is to install a copper thimble between the inner and outer thimbles and apply a thin layer of thermal grease in the gap between the thimbles to ensure good contact, hereinafter referred to as the copper-medium FBG sensor, the heat transfer method is also heat conduction. As shown in Figure 3, six double-thimble-type FBG temperature sensors were manufactured and encapsulated by three materials as mentioned above. The FBG temperature sensor is designed with two FC/APC connectors that can meet the scalability of the measurement system and distributed temperature measurement requirements.

### 2.2. Theoretical Derivations of Temperature Sensitivity

Ideally, the applied stress should not affect the wavelength shift of the fiber and the bonding method of the fiber should isolate the apparent thermal strain of the metal thimble. The wavelength shift should only be related to the ambient temperature [20]. Under thermal load, the outer thimble is in contact with the ambient temperature, and the heat is transferred to the inner thimble through the material of the filling layer, and finally, transferred to the FBG. Besides, the temperature will make the fiber Bragg grating produce thermal expansion, thermo-optic effect, and other parameters [7,21,22]. Expanding Equation (Equation 2), the relative wavelength shift of FBG caused by temperature is
(4)$$\Delta \lambda_{B}=2\left[\frac{\partial n_{eff}}{\partial T}\Delta T+{\left(\Delta_{neff}\right)}_{ep}+\frac{\partial n_{eff}}{\partial a}\Delta a\right]\Lambda +2_{neff}\frac{\partial \Lambda }{\partial T}\Delta T,$$
where, $\alpha_{n}=\left(1/n_{eff}\right)\partial n_{eff}/\partial T$ is the thermo-optical coefficient, ${\left(n_{eff}\right)}_{ep}$ is the elastic-optical effect caused by thermal expansion, $\partial n_{eff}/\partial a$ is the waveguide effect caused by the change of the fiber core diameter caused by thermal expansion, and $\alpha_{\Lambda }=\left(1/\Lambda \right)\partial \Lambda /\partial T$ is the linear thermal expansion coefficient of the optical fiber. Equation (Equation 4) can be rewritten as
(5)$$\frac{\Delta \lambda_{B}}{\lambda_{B}\Delta T}=\frac{1}{n_{eff}}\left[n_{eff}\alpha_{n}+{\left(\Delta_{neff}\right)}_{ep}+\frac{\partial n_{eff}}{\partial \alpha }\frac{\Delta \alpha }{\Delta T}\right]+\alpha \Lambda .$$

Therefore, the temperature-induced strain is
(6)$$\left[\begin{array}{c} \varepsilon_{rr}\\ \varepsilon_{\theta \theta }\\ \varepsilon_{zz} \end{array}\right]=\left[\begin{array}{c} \alpha \Delta T\\ \alpha \Delta T\\ \alpha \Delta T \end{array}\right].$$

Neglecting the waveguide effect, the temperature sensitivity coefficient of the fiber grating is
(7)$$K=\frac{\Delta \lambda_{B}}{\lambda_{B}\Delta T}.$$

### 2.3. Dynamic Heat Transfer Process

The temperature response of the sensor is related to transient heat conduction. The temperature at any point in the sensor is a general nonlinear function of time, and the temperature increases or decreases over time and gradually tends to a certain temperature. Time constant quantifies the time that the sensor needs to react to a sudden temperature variation. Response speed is generally characterized by the time constant τ, which corresponds to the time needed for the temperature sensor output signal to reach 63.2% of the steady value after a step temperature jump. The FBG temperature sensor can be regarded as a multilayer composite cylindrical wall structure, the temperature of the wall is uniform and has no internal heat source. Suppose that the experimental process does not damage the system’s heat balance. When the ambient temperature is higher than the sensor axis temperature, the heat is conducted from external to the FBG. According to the structure diagram of the FBG temperature sensor, the time constant of each sensor is divided into three parts: The first part of the time constant is $\tau_{1}$, the time for heat to transmit from the outside to the stainless steel thimble. The second part $\tau_{2}$ is the time for heat to transmit from the filling layer to the copper thimble, and $\tau_{3}$ is the time from the copper to the grating. The total time constant is the sum of $\tau_{1}$, $\tau_{2}$, and $\tau_{3}$. When the ambient temperature is lower than the sensor axis temperature, the heat is conducted from the axis to the external. Both cases change $\lambda_{B}$, such that the temperature of the ambient can be measured.

In the process of transient heat conduction, the internal energy of objects in different positions changes with time, which leads to the difference in the amount of heat conducted. For the previous FBG temperature sensor with a multilayer circular thimble composite wall structure, the amount of heat at different positions in the heat transfer direction is not equal [23]. Take the stainless steel thimble as an example for analysis, the change in the amount of heat conduction of the internal and external contact surfaces at different times is qualitatively shown in Figure 4, where $\Phi_{1}$ is the heat introduced into the stainless steel thimble from the outside and $\Phi_{2}$ is the heat exported from the stainless steel thimble to the thermal grease. The two heat fluxes are not equal during the whole transient-state heat conduction process, but as the process progresses, the difference gradually decreases until the two heat fluxes reach equilibrium in the steady-state stage. The shaded area in Figure 4 represents the energy accumulated by the stainless steel thimble during the heating process.

### 2.4. Theoretical Derivations of Temperature Sensor Dynamic Response

According to the relationship between the temperature change characteristics of the object and the boundary condition parameters under the third type of boundary conditions, place the sensor in a fluid with a temperature of $t_{\infty }$ for heating; the heat transfer coefficient of the contact surface is *h*, and the thermal conductivity of the thermal thimble is λ. According to the difference in the relative size of the thermal-conduction resistance (δ/λ) of the circular thimble and the surface contact heat transfer resistance (l/h), the change of the temperature field in the circular thimble will have different situations. When l/h<<δ/λ, since the surface contact heat transfer resistance is almost negligible, the surface temperature of the thimble is heated to $t_{\infty }$ at the beginning of the process. As time goes by, the temperature of the thimble gradually rises and tends to $t_{\infty }$ [24]. The ratio of the internal thermal-conduction resistance to the surface contact heat transfer resistance of a thermally conductive object is called the characteristic number $B_{i}$
(8)$$B_{i}=\frac{l/\lambda }{1/h}=\frac{lh}{\lambda },$$
where *l* is the characteristic size, in the composite wall of multilayer round pipe, when V/A=l is selected as the characteristic size, $B_{i}$ is recorded as $B_{i}v$; V is the volume; and A is the heat transfer surface area of the heat transfer structure. When $B_{i}\to 0$, the heat absorbed by the heat exchange surface is instantly evenly distributed among the various parts of the object, and the temperature varies uniformly and synchronously everywhere. Such a heat conduction system is called an LHC system [25,26]. The temperature of the LHC system is only a unitary function of time and is independent of the spatial coordinates. Suppose that the mass and heat capacity of the object originally continuously distributed are concentrated at one point, with only one temperature value. This simplified analysis method that ignores the internal heat conduction and thermal resistance of the object is called the lumped parameter method [27,28].

In the sensor measurement system, it is assumed that the contact heat transfer coefficient between the two objects in contact and the solid thermophysical properties remain constant [29]. The structural dimensions, thermophysical properties, and $B_{i}$ of the double-thimble FBG temperature sensor are shown in Table 1. For measuring ambient temperature with the FBG temperature sensor, when the order of magnitude of $B_{i}$ number is less than 0.001, the lumped parameter method can be used to analyze the sensor temperature field, and sufficient and accurate solutions can be obtained to meet the measurement requirements. According to the influence of $B_{i}$ on the transient temperature distribution of an object and the characteristics of LHC system, the transient heat conduction mathematical model is established for each layer of the thimble wall of the sensor. It is worth noting that the $B_{i}$ number is larger than 0.001 when the filling material is air medium, but the radial dimension of the layer is very small, so it is still regarded as a LHC system.

The thermal conductivity of the inner thimble, outer thimble, and filling layer of the sensor is quite different, so, the sensor is regarded as a multicapacity system composed of three LHC systems. According to the one-dimensional heat conduction model and the energy balance relationship of the third type of boundary conditions, the temperature response of a single heat capacity system is solved separately. For the object without an internal heat source, the net heat flow introduced into the control volume is equal to the increase in the internal energy of the substance in the control volume [11,30]. Take the stainless steel thimble as an example, the transient heat conduction differential equation is
(9)$$\rho Vc_{p}\frac{dT}{dt}=-\int_{A}q\cdot ndA=\int_{A}h\left(T-T_{i}\right)dA,$$
where the left side of the formula is the increase in the internal energy of the outer thimble, q represents the heat flux density of the outermost thimble surface, and n represents the unit external normal vector on the surface of the thimble. $T_{i}$ is the initial temperature, *T* is the temperature at time *t*; introduce the intermediate variable $\theta =T-T_{i}$ and separate the variables of Equation (Equation 9) to get
(10)$$\frac{d\theta }{\theta }=\frac{-\int_{A}hdA}{\rho Vc_{p}}dt.$$

Then, consider the initial conditions $\theta_{0}=T_{0}-T_{i}$ and the characteristic number $B_{i}$ in Equation (Equation 10) with respect to *t*, from 0 to *t*, to get
(11)$$\int_{\theta_{0}}^{\theta }\frac{d\theta }{\theta }=-\int_{0}^{t}\frac{h^{2}}{B_{iv}\rho c_{p}\lambda }dt$$
(12)$$\Theta =\frac{\theta }{\theta_{0}}=\frac{T-T_{i}}{T_{0}-T_{i}}=exp\left(-\frac{h^{2}}{B_{iv}\rho c_{p}\lambda }\cdot t\right).$$

The temperature can be solved by
(13)$$T=T_{i}+\theta_{0}\frac{T-T_{i}}{T_{0}-T_{i}}=T_{i}+\theta_{0}exp\left(-\frac{h^{2}}{B_{iv}\rho c_{p}\lambda }\cdot t\right).$$

When the lumped parameter model is used to analyze the temperature response of the thimble, the excess temperature Θ in the thimble changes exponentially with time. The temperature changes rapidly at the beginning of the process and then gradually slows down, as shown in Figure 5.

In the exponential function Equation (Equation 12), $h^{2}/B_{iv}\rho c_{p}\lambda$ and 1/t have the same dimensions; when $t=B_{iv}\rho c_{p}\lambda /h^{2}$, T reaches 63.2% of the temperature jump, that is, τ=t is the time constant of the sensor. The time constant depends on the sensor’s characteristic number ($B_{iv}$) determined by characteristic size, the material’s thermophysical properties ($\rho ,c_{p},\lambda$), and the surface heat transfer conditions (*h*). The larger the heat capacity, the slower the temperature change; the better the surface heat transfer conditions, the more heat can be transferred per unit time, so that the sensor temperature can quickly approach the temperature of the measured object. The double-thimble-type FBG temperature sensor is a multicapacity system composed of multiple single-capacity systems. The structural dimensions and thermophysical parameters of each part of the sensor are substituted into the heat transfer model, the time constant of the sensor can thus be obtained. The time constant of the air-medium FBG temperature sensor, grease-medium FBG temperature sensor, and copper-medium FBG temperature sensor are 116 s, 146 s, and 78 s, respectively.

## 3. Numerical Simulation and Experiments

### 3.1. Simulation of Temperature Sensor Dynamic Response

The analysis method of the temperature response of a single-capacity system is, in principle, applicable to each LHC system. However, there are certain errors in the calculation model of the sum of multiple single-capacity systems. Building an energy equations set consisting of multiple single-capacity systems to analyze the temperature response of the multicapacity system is too cumbersome to solve the solution process, and the expression of the solution is too complicated, which is not convenient for quantitative calculations to obtain digital results. For multicapacity systems with complex boundary conditions and variable physical properties, the numerical solutions based on discrete mathematics can be used. This method replaces the original continuously distributed temperature field with approximate temperature values at a finite discrete point in space or space and time, and establishes and solves the algebraic equations about the temperature values of these nodes in a certain way, that is, the approximate temperature field is the obtained solution. One of them, the finite element analysis method, can transform these partial differential equations into a set of algebraic equations on the solution area so that the approximate solution can be obtained by computer solving [9,31,32].

Based on the FBG sensor structures proposed above, a proportional three-dimensional transient heat transfer simulation model of the sensor is established, as shown in Figure 6. Corresponding thermophysical parameters for materials with different structures were set, the element type is defined as a three-dimensional ten-node tetrahedral element, the structure is divided into 5,292,466 grids, and the heat exchange surface is finely divided into 174,364 grids. The boundary conditions to be solved are as follows:The fiber grating, outer thimble, filling material, and inner thimble are considered as isotropic materials, and their thermophysical parameters are constant. The material characteristics are shown in Table 1.The outer thimble surface of the sensor is in contact with air, and it is set as a fluid–solid interface coupling surface, and the natural convection heat transfer coefficient of air is set to 20 $W/m^{2}K$. The interface between the sensor surface and the air is set as a nonslip boundary condition.To calculate the time required for the response process of the sensor by setting the ambient temperature to 80 ${}^{{}^{\circ}}$C and the sensor temperature to 20 ${}^{{}^{\circ}}$C, a solution is performed every 0.2 s to calibrate the sensor.To calculate the time required for the recovery process of the sensor by setting the ambient temperature to 20 ${}^{{}^{\circ}}$C and the sensor temperature to 80 ${}^{{}^{\circ}}$C, a solution is performed every 0.2 s to get the heat dissipation process of the sensor.

The dynamic response of the sensor temperature field and the fiber grating temperature at different moments can be obtained by solving the heat conduction process. When the ambient temperature jumps positively from 20 to 80 ${}^{{}^{\circ}}$C, as shown in Figure 7a, the time constants of positive temperature jumps of the air-medium FBG sensor, grease-medium FBG sensor, and copper-medium FBG temperature are 109 s, 137 s, and 190 s, respectively. When the ambient temperature jumps negatively from 80 to 20 ${}^{{}^{\circ}}$C, as shown in Figure 7b, the time constant is 123 s, 149 s, and 209 s, correspondingly. So, the time constant of positive temperature jump is shorter than time constant of negative temperature jump, response speed depends on whether the sensor is heated or cooled, and packaging materials with different thermal conductivity have a greater impact on response speed. In the heating or cooling process, the temperature response speed of each point inside the sensor is also different. As shown in Figure 8a,b, the difference of the response speed between the surface and the axis of the air-medium FBG sensor is nearly 40 s; however, for the grease-medium FBG sensor and the copper-medium FBG sensor, the temperature response speed of surface and the axis have only a 2-s difference in the heating process shown in Figure 8c,d.

### 3.2. Experiment of FBG Temperature Sensitivity

Based on the above theoretical analysis, experiments are then performed to investigate the temporal response of the three medium types of sensors in temperature sensing with the test experimental apparatus shown in Figure 9. An amplified spontaneous emission (ASE) broadband light source (with an optical spectrum from 1515∼1595 nm and output power of 10 mW) was used. The resolution of the optical signal demodulation equipment is 1 pm, the sampling frequency is 2 kHz, and the wavelength demodulation range matches the ASE broadband light source. In the experiment, a temperature and humidity programmable chamber (Temp-Humi PC) is used to provide temperature jumps, the controllable temperature range is −30 ${}^{{}^{\circ}}$C∼120 ${}^{{}^{\circ}}$C, and the accuracy is 0.01 ${}^{{}^{\circ}}$C, which meets the requirements of the experimental system [3,11]. In the experiment, the broadband light output by the ASE is injected into the FBG temperature sensor through the circulator, the temperature sensor grating returns the light signal of a specific wavelength, and the reflected light enters the demodulation device through the circulator. The demodulation device detects and outputs the center wavelength of the maximum back-reflected light, and the host computer converts the wavelength data into a temperature.

When calibrating the static performance of the FBG temperature sensors, connect the sensors in series and place them in the Temp-Humi PC, set a reference temperature between 0 and 100 ${}^{{}^{\circ}}$C at a temperature interval of 10 ${}^{{}^{\circ}}$C, the ambient humidity is set to 30%Rh and remains constant. After the Temp-Humi PC reaches the reference temperature and stabilizes for a period of time, the center wavelength corresponding to each reference temperature is recorded, repeat the heating and cooling cycle five times, thereby, the calibration curve of the FBG temperature sensor is obtained.

Figure 10 shows the relationship curve between the wavelength shift and the temperature change of the air-medium FBG sensor, and the curve between the wavelength shift and grease-medium FBG sensor. Equations (Equation 14) and (Equation 15) are calibration curves of the the air-medium and grease-medium FBG sensor obtained by least square fitting,
(14)$$\lambda_{a}=0.0186\times 10^{-3}T+1538.548$$
(15)$$\lambda_{g}=0.01031\times 10^{-3}T+1544.603.$$

The sensitivity coefficients of the air- and the grease-medium FBG sensor are 0.0186 nm /${}^{{}^{\circ}}$C (18.6 pm/${}^{{}^{\circ}}$C) and 10.3 pm/${}^{{}^{\circ}}$C, the R-Square (COD) of the sensors can reach 0.99875 and 0.99978, and the nonlinear error of the sensors are 1.7% and 0.9%, respectively. The calibration data of the copper-medium FBG sensor is fitted by the same method, the sensitivity coefficient is 10.6 pm/${}^{{}^{\circ}}$C, the R-Square (COD) is 0.99976, and the nonlinear error is 1.1%. It can be seen that the nonlinear errors of the double-thimble FBG temperature sensor filled with thermal grease and copper thimble are decreased compared with the unfilled structure.

### 3.3. Experiment of Response Time

There is no forced air circulation in the Temp-Humi PC, it was considered that the sensors were exposed to natural convection. The temperature jump ($T_{\infty }$) is provided by the programmable chamber and the initial temperature is the ambient temperature ($T_{i}$) of the thermostatic room at 20 ${}^{{}^{\circ}}$C. First, set a positive temperature jump ($T_{+\infty }$) to 80 ${}^{{}^{\circ}}$C, when the Temp-Humi PC reaches 80 ${}^{{}^{\circ}}$C, the three type FBG temperature sensors are put into the Temp-Humi PC at the same time by using the automatic input device controlled by the electromagnet; meanwhile, wavelength data is collected and recorded by the FBG demodulator instrument in real-time. Wait for the temperature inside and outside the sensor to reach a new balance, that is, the fluctuation range of the reflection center wavelength reaches within the error range. Then, take the sensors out of the Temp-Humi PC and place them in the thermostatic room at 20 ${}^{{}^{\circ}}$C.

The wavelength shift data is converted into temperature data through the wavelength-temperature calibration curve, and the dynamic response curves of the sensors were obtained as shown in Figure 11. In the heating process, the time constant of the air-medium FBG sensor, the grease-medium FBG sensor, and the copper-medium FBG temperature is 100 s, 136 s, and 153 s, respectively; in the cooling process, the time constant is 120 s, 196 s, 214 s, respectively. The results obtained show that response speed is faster during the heating phase than the cooling phase, and packaging materials with different thermal conductivity have a great impact on the dynamic response. The response speed and recovery speed of air-medium FBG sensor is faster than grease-medium and copper-medium FBG temperature sensors.

## 4. Discussion

The time constant of double-thimble FBG temperature sensors with three package structures obtained by different research methods is shown in Table 2.
In the production of the copper-medium FBG sensor, there is a certain gap between the copper thimble and the inner thimble, and between the copper thimble and the outer thimble. There is a large contact thermal resistance on these contact surfaces, and the heat transfer is greatly affected. Based on this, the established equal-scale three-dimensional simulation model also has two gaps. However, this structure is not considered in the transient heat conduction mathematical model, the ideal contact model (the copper is in good contact with the inner and outer thimble) is applied, so the time constant obtained by the mathematical model is smaller than that of the other two methods.It can be seen that the time response of the sensor is related to the heat storage capacity and thermal conductivity of the packaging material. In different research methods, the response speed of the air-medium FBG temperature sensor is more than 20% faster than that of the grease-medium FBG sensor, and more than 30% faster than that of the copper-medium FBG sensor. The smaller the heat storage capacity and the larger the heat transfer coefficient, the faster the response of the sensor.In the heating process, the system has an infinite external heat source, but in the cooling process, the heat source is the finite heat absorbed by the sensor. So, in the heating and cooling process, the difference in heat density causes the time constant is shorter during the heating phase than the cooling phase.For the same encapsulation structure, the response and recovery times obtained by different research methods are consistent and the deviation is within the acceptable range (except the copper-medium FBG sensor), indicating that the analytical solution and finite element simulation of the transient heat conduction model have certain engineering application value.The model of the sensor is based on an air bath, that is, the heat exchange method with the external environment is natural convection heat exchange. In actual engineering applications, the environment the double-thimble FBG temperature sensor is exposed to is water or concrete with a larger contact heat transfer coefficient, so the sensor has a faster response speed.

## 5. Conclusions

Response speed is a vital parameter when selecting a temperature sensor, as underestimating the time constant can lead to significant measurement errors. However, despite its importance, to date, there has been no generally accepted method of obtaining the response speed of FBG temperature sensors. In this paper, a transient heat conduction solution model is built and the time constant is solved analytically. Moreover, a numerical model of the temperature sensor is built and the finite element simulation is carried out to obtain the temperature field. The double-thimble-type of FBG temperature sensor is designed and fabricated, and the dynamic characteristics of the sensors are tested. Combining the package structure and experimental data, the influencing factors of the sensor’s dynamic response characteristics are analyzed. The results indicate the following:The response speed depends on the thermal properties of the sensor encapsulation materials.The time constant depends on whether the sensor is heated or cooled, heating time response for the sensor is shorter than the cooling time response.The dynamic performance of the sensor can be verified quickly and effectively by establishing a transient differential equation and finite element model based on heat balance.This work can provide inspections for the trial manufacture and dynamic calibration of FBG temperature sensors.

## Figures and Tables

**Figure 1 micromachines-12-00016-f001:**
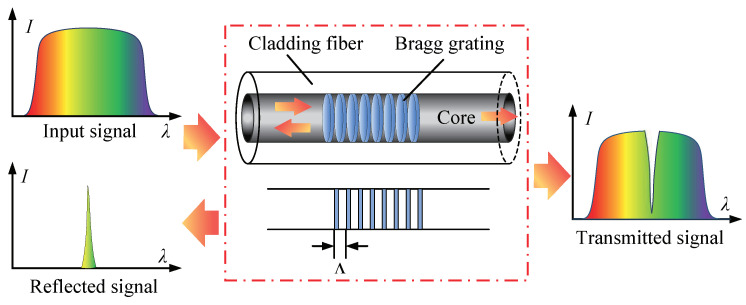
Transmission and reflection spectrum of fiber Bragg grating (FBG).

**Figure 2 micromachines-12-00016-f002:**
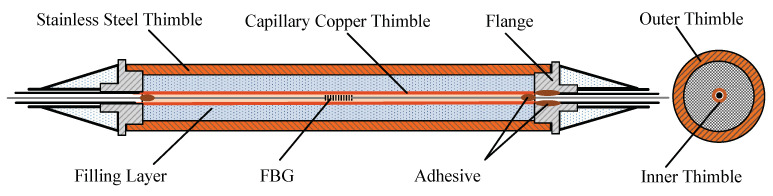
The structure diagram of the FBG temperature sensor.

**Figure 3 micromachines-12-00016-f003:**
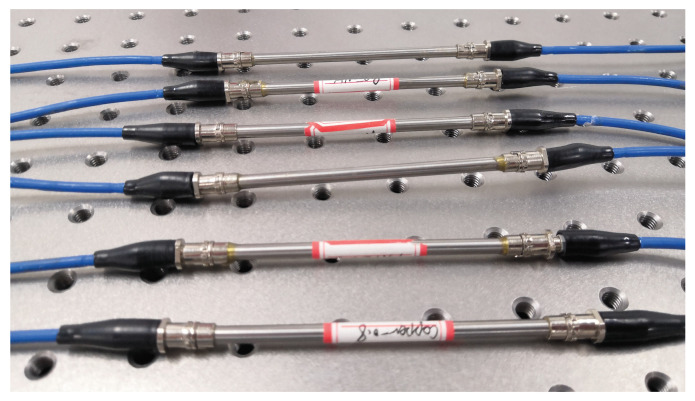
Double-thimble-type FBG temperature sensors developed in the laboratory.

**Figure 4 micromachines-12-00016-f004:**
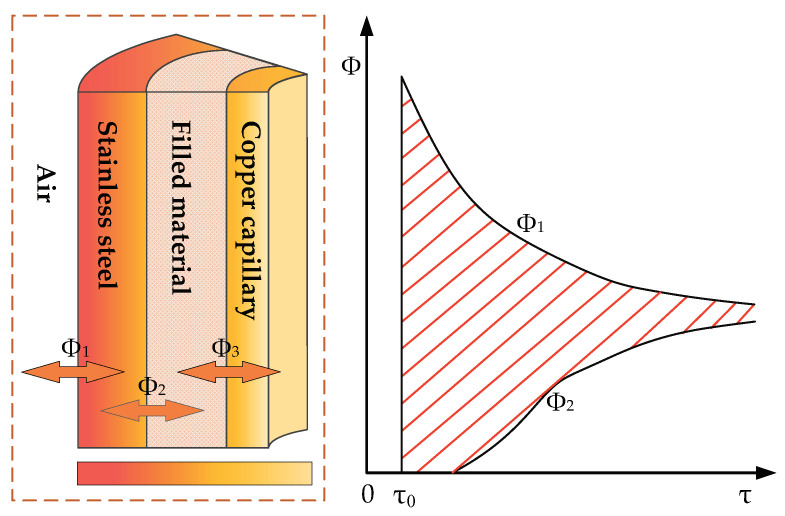
The amount of heat conduction on both sides of the cylinder with time during the process of transient heat conduction.

**Figure 5 micromachines-12-00016-f005:**
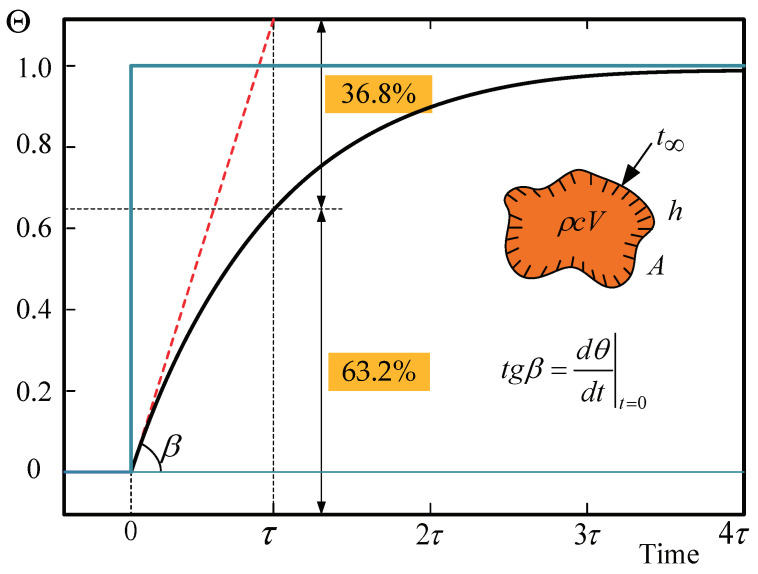
Dimensionless excess temperature of the object changes with time.

**Figure 6 micromachines-12-00016-f006:**
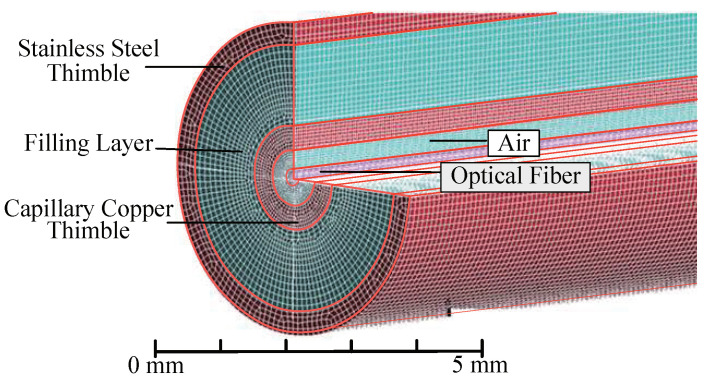
Finite element meshing diagram of FBG temperature sensor transient heat transfer model.

**Figure 7 micromachines-12-00016-f007:**
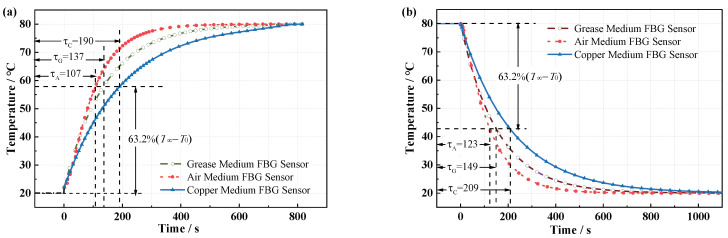
(**a**) Time response of positive temperature jump for FBG temperature sensor calculated by simulation. (**b**) Time response of negative temperature jump for FBG temperature sensor.

**Figure 8 micromachines-12-00016-f008:**
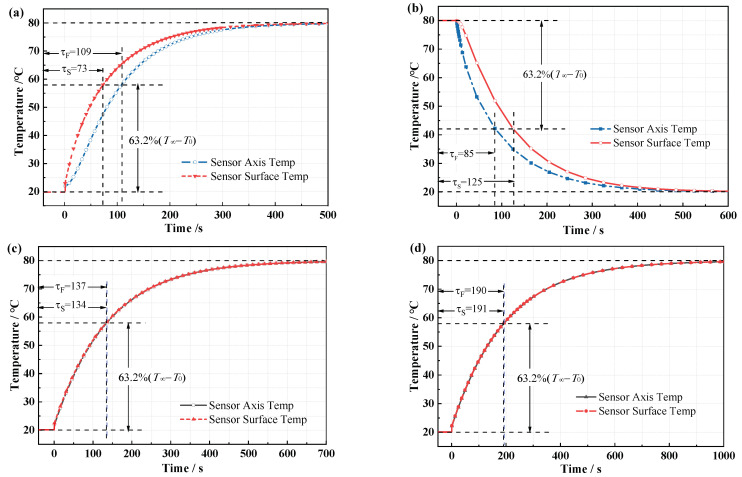
(**a**) Time response of positive temperature jump for the air-medium FBG temperature sensor’s surface and axis. (**b**) Time response of negative temperature jump for the air-medium FBG temperature sensor’s surface and axis. (**c**) Time response of positive temperature jump for the grease-medium FBG temperature sensor’s surface and axis. (**d**) Time response of positive temperature jump for the copper-medium FBG temperature sensor’s surface and axis.

**Figure 9 micromachines-12-00016-f009:**
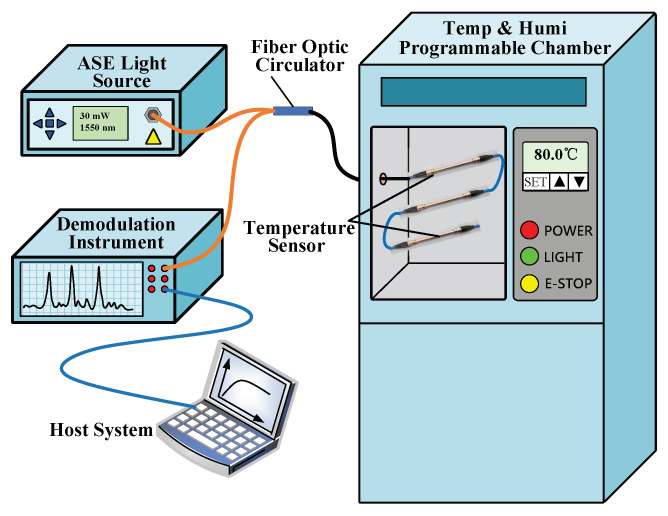
The experimental apparatus of the FBG temperature sensors.

**Figure 10 micromachines-12-00016-f010:**
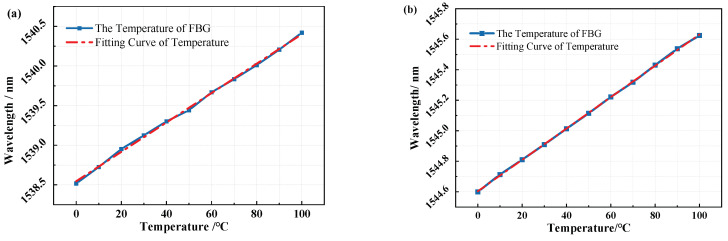
(**a**) Calibration curve of the air-medium FBG temperature sensor, (**b**) Calibration curve of the grease-medium FBG temperature sensor.

**Figure 11 micromachines-12-00016-f011:**
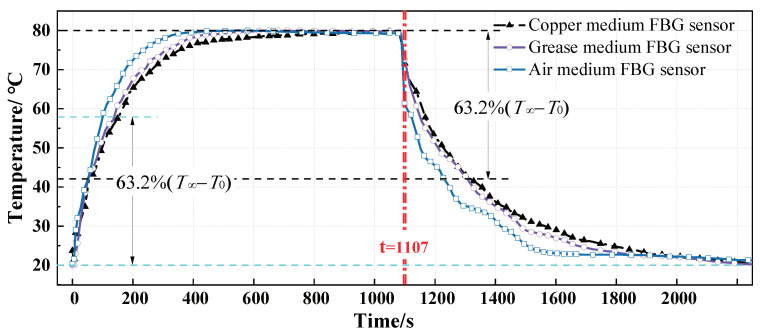
The temperature curve at the axis obtained by the experimental method.

**Table 1 micromachines-12-00016-t001:** Structural dimensions and thermophysical parameters of the double-thimble FBG temperature sensor.

Physical Parameters	Outer Thimble	Inner Thimble	Filling Layer		
	**304# Stainless Steel**	**Red Copper**	**Air**	**Thermal Grease**	**Red Copper**
$V/m^{3}$	3.46 $\times 10^{-7}$	1.01 $\times 10^{-7}$	7.85 $\times 10^{-7}$	7.85 $\times 10^{-7}$	7.85 $\times 10^{-7}$
$A/m^{3}$	1.13 $\times 10^{-3}$	3.77 $\times 10^{-4}$	9.65 $\times 10^{-4}$	9.65 $\times 10^{-4}$	9.65 $\times 10^{-4}$
$\lambda /W\cdot m^{-1}C^{-1}$	16.2	401	0.026	10	401
$\rho /\mathrm{kg}\cdot m^{-3}$	8055	8900	1.1614	2013	8900
$c_{p}/J\cdot {\mathrm{kg}}^{-1}\cdot C^{-1}$	480	385	1007	850	385
$B_{iv}$	4.67$\times 10^{-4}$	1.34 $\times 10^{-5}$	6.26 $\times 10^{-2}$	6.78 $\times 10^{-4}$	4.06 $\times 10^{-5}$

**Table 2 micromachines-12-00016-t002:** The time constant of the sensor calculated by different research methods.

Type of FBG Sensor	Mathematical Model/s	Simulation Results/s		Experiments Results/s	
		$T_{+\infty }$	$T_{-\infty }$	$T_{+\infty }$	$T_{-\infty }$
Air	116	109	123	100	120
Grease	146	137	149	136	196
Copper	78	190	209	153	214

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
