# Peer review of "Analytical Evaluation and Experiment of the Dynamic Characteristics of Double-Thimble-Type Fiber Bragg Grating Temperature Sensors"

_micromachines, 2020, doi:10.3390/mi12010016_

Round 1
Reviewer 1 Report
Firstly, I would like to thank you for giving me the opportunity to review your research work. I have carefully read the paper titled "Analytical evaluation and experimental of the dynamic characteristics of double-thimble type FBG
temperature sensors".
I am of the opinion that it is well structured and in general easy to understand.
However, there are some points, which should be addressed and described in more details:
1. Please give more comments why the response time of FBG temperature sensors is varying between the mathematical model, simulation, and experimental results experiments (Table 2).
2. Which method is used in the demodulation instrument for peak detection of FBG temperature sensor’s reflected signal? How the precise position of the peak is determined (page 10)?
3. What are the main reasons why the coefficients of FBG temperature sensor (air and copper medium 10.3 pm/°C, but copper medium 10.6 pm/°C) are variating (page 11)?
4. Authors have used ASE as a light source. What are the reasons why the ASE was chosen as the broadband light source? Why not the broadband SLED?
5. Please check and correct % value interval in the abstract: “The results show that the dynamic response performance depends on the encapsulation material parameters, the response time is shorter than recovery time, the response speed of the air packaging sensor is 20.4% 26.4% faster than the grease packaging sensor, and 34.6% 42.6% faster than the copper packaging sensor.”
6. The caption of figure 5 should be fixed. Please specify what is seen in the model. Is it a piece of fiber or fiber with inscribed FBG? If it is a FBG, we should see a grating structure in the core. Please indicate where are the core and cladding of the fiber.
Also, please check that all abbreviations are explained when they first appear in the text (e.g. FBG, RFBG, ASE, etc.)
Author Response
Response to Reviewer 1 Comments
Thank you for your comments concerning our manuscript entitled Analytical evaluation and experimental of the dynamic characteristics of double-thimble type FBG temperature sensors (Manuscript ID: micromachines-1039508). Those comments are all valuable and very helpful for revising and improving our paper, as well as the important guiding significance to our researches. We have studied comments carefully and have made correction which we hope meet with approval. Revised portion are marked in red in the paper. The main corrections in the paper and the responds to the reviewer’s comments are as flowing:
Point 1. Please give more comments why the response time of FBG temperature sensors is varying between the mathematical model, simulation, and experimental results experiments (Table 2).
Answer: Answer: For different packaging structures, the dimensions of the inner thimble and outer thimble are the same, and the thermal physical parameters of the inner thimble and outer thimble material are exactly the same. so the difference in response speed of the sensor comes from two aspects: 1, the thermal physical properties of the filling material; 2, the heat transfer between the filling layer and the inner and outer thimble characteristic. In different environments, the heat transfer coefficients of enclosed air and grease are quite different. We consider the air flow state and the filling state of the thermal grease (the thermal grease is not in good contact with the casing), so the heat transfer coefficient of the thermal grease is estimated to be about 300 times that of air. The thermophysical parameters of the packaging material and the sensor geometry are taken into the time constant formula, and it is obtained that the time constant when filled with grease is approximately 2.5 times that of air. Therefore, the time constant of the grease medium sensor is larger than that of air.
For calculating the time constant of the copper medium sensor, the contact state between copper and stainless steel, and between copper and copper cannot be estimated, and the heat transfer coefficient cannot be converted equivalently. So the ideal model is used (assuming that the copper is in good contact with the inner and outer thimble). However, in the experiment, there are two gaps between the copper and the inner, and between the copper and the outer thimble. The interface has a larger contact thermal resistance, and the interface heat transfer coefficient becomes smaller. The simulation model is based on the experimental structure and size. So the response speed of copper is the best in theoretical calculations, while in the modeling and experimental results, the response speed of copper has deteriorated.
(line 331-334, marked in red)
Point 2. Which method is used in the demodulation instrument for peak detection of FBG temperature sensor’s reflected signal? How the precise position of the peak is determined (page 10)?
Answer: There is an optical signal analysis module in the demodulator, which includes a spectrum analysis unit, a detection unit and an electrical processing unit. The spectrum analysis unit separates the wavelength of the incident optical signal, the detection unit converts the optical signal into an electrical signal, and the electrical processing unit processes electrical signals containing spectral information. The spectrum reflected or transmitted by the grating has a large energy difference at different wavelengths. Through high-order fitting of the electrical signal of the spectrum information, the center wavelength position of the reflected signal can be accurately identified. The reflection signal peak detection method is integrated in this module and is provided by the manufacturer. This article only uses the demodulation module, and does not do specific research on the wavelength detection method.
Point 3. What are the main reasons why the coefficients of FBG temperature sensor (air and copper medium 10.3 pm/°C, but copper medium 10.6 pm/°C) are variating (page 11)?
Answer: The temperature sensitivity coefficient of the grating is affected by the production process. Even the same batch of bare fibers produced by the same factory, the characteristics are different. In the experiment, the optical fiber pasted in the copper capillary tube is slightly longer than the copper tube, so that the fiber grating is not affected by the apparent thermal stress, that is, the grating has no sensitization and desensitization structure,and the coefficient is irrelevant to the packaging material. We revised the content of the temperature sensitivity coefficient. In addition, in the experiment, the nonlinear error is one of the important characteristics of the temperature sensor, and we found that different packaging materials have a significant effect on the nonlinear error of the sensor, so this part of the content is added to the manuscript.
(line 301-307, marked in red)
Point 4. Authors have used ASE as a light source. What are the reasons why the ASE was chosen as the broadband light source? Why not the broadband SLED?
Answer: In fact, both amplified spontaneous emission (ASE) and super luminescent LED (SLED) can be used as the light source of the experimental system. The output power of the ASE light source is 10 mW, and the power stability is high. The optical spectrum is from 1515 to 1595 nm, and it has good flatness. The ASE light source fully meets the requirements of the test system. Of course, SLED light source has more flexible wavelength selection and wider wavelength coverage. It is an excellent light source equipment and is also suitable for this system.
Point 5. Please check and correct % value interval in the abstract: “The results show that the dynamic response performance depends on the encapsulation material parameters, the response time is shorter than recovery time, the response speed of the air packaging sensor is 20.4% 26.4% faster than the grease packaging sensor, and 34.6% 42.6% faster than the copper packaging sensor.”
Answer: we have re-written this sentence (line 11-13, marked in red).
Point 6. The caption of figure 6 should be fixed. Please specify what is seen in the model. Is it a piece of fiber or fiber with inscribed FBG? If it is a FBG, we should see a grating structure in the core. Please indicate where are the core and cladding of the fiber.
Answer: Figure 6 shows the finite element meshing diagram of the transient heat transfer model for the FBG temperature sensor. As shown in the figure, the model including stainless steel thimble, filling layer, capillary copper thimble, and optical fiber are respectively marked in the figure. Because the fiber diameter is much smaller than the diameter of the package structure, the temperature of the grating is similar to that at the sensor’s axis, therefore, the cladding, fiber core, and fiber grating are not subdivided. In addition, we have also modified the title of the figure.
We have re-edited the Fig. 6 to meet the journal's requirements. (in Page 9, marked in red).
Point 7. Also, please check that all abbreviations are explained when they first appear in the text (e.g. FBG, RFBG, ASE, etc.)
Answer: We checked the abbreviations that appeared in the text and revised them according to the specifications.

Reviewer 2 Report
Review of “Analytical evaluation and experimental of the dynamic characteristics of double-thimble type FBG temperature sensors”
The results are interesting and informative. The authors propose an engineering challenge in effectively encapsulating this type of sensor to create a robust structure without compromising the response time. Three filling materials are tested: air, silicon grease, and “red copper” material.
One of the main conclusions that the authors report is that “The smaller the heat storage capacity and the larger the thermal conductivity, the faster the response of the sensor.” This seems to partially agree with the analytical solution presented by the authors, where the copper fill presents a faster response than the other two, although one would expect that thermal grease, with a heat conductivity 2000 times that of air would respond faster. The modeling and experimental results entirely contradict the analysis and the conclusion, but no clear explanation for this is given by the authors. This is surprising and contradictory. Please explain this.
The authors pose the main question that sensor robustness through encapsulation and fast response times are contradictory. However, the experiments only focus on sensor response. What is the effect of the different studied encapsulations materials in the robustness of the sensor? It seems like this information is necessary if the sensor response is only improved by a few seconds but the robustness is compromised much more significantly. Additional experiments may not be needed but justification may be helpful.
Time constant and response time are used by the authors indistinctly. I would argue that the response time would be the time to get to 99.5 of the final value (5*tau), while the time constant would be to 63.2% (1/e or tau).If a different definition is presented, please justify.
Briefly describe in the introduction what FBG temperature sensors are and how they work so that the readers can put this study in context and better understand the challenges presented by the authors.
What follows are just some grammatical fixes that could help improve the quality of the paper. This list is not exhaustive; I recommend extensive review of grammar.
Line 18 - FBG temperature sensors “are” instead of “were”
Line 22-23 – Does the temperature sensor have excellent mechanical strength? Or is it the packaging that provides excellent mechanical strength? Please modify the sentence to clarifty.
Line 24 - in order to achieve response (not respond) to the ambient temperature rapidly
Line 32 – “A few researchers” or “research studies”
Line 34 – delicate instead of delicat
Line 36 – what is an RFBG sensor?
Line 37 and throughout – Place a space between numbers and units (i.e 9 s instead of 9s)
Lines 51-53 – incomplete and run-on sentence, please re-write
Line 67 – period should be replaced by a comma.
Line 54 – Uses three types of materials
Line 55 – what does “respectively” refer to at the end of the sentence?
Line 64 – Did the authors fabricate the sensors? Or where they purchased commercially? This is not clear. If the focus on the paper is also on the grating structure it may be helpful to include a schematic explaining how the device works in relation to equation 1, 2 and 3.
Line 66-69 – Please fix the punctuation of these three sentences. It is difficult to understand the meaning.
Line 71- “… , which ultimately affect …”
Paragraph after Line 72 – This is said twice: “The capillary has good thermal conductivity and flexibility”
It is not clear to me what the “reserved length” is. Please explain.
Line 80 – In contrast, an encapsulation ….
Line 86 – I fail to see the relevance of this sentence in lines 84-86. Please, help the reader make the connection and clarify how this relates to an air medium: “According to the Grashof criterion [19], fluid flow is restricted in a limited interlayer space and the development of fluid thermal boundary layer is disturbed, when Grd 2430, the heat transfer method of fluid is heat conduction.”
Line 97 – Ideally, the applied stress should have no effect on the wavelength shift of the fiber, and the bonding method of the fiber should isolate the apparent thermal strain of the metal thimble. The wavelength shift should only related to the ambient temperature [20].
Line 103 and others – The correct abbreviation for Equations should be Eq. and not Equ.
Line 108-111 – Please rewrite sentence, it is difficult to understand the meaning.
Line 120 – Both cases change lB , such that the…
Line 122 – The word because should not be capitalized. The sentence is long and difficult to understand, please break up in smaller ones.
And many more….
Author Response
Response to Reviewer 1 Comments
Thank you for your comments concerning our manuscript entitled Analytical evaluation and experimental of the dynamic characteristics of double-thimble type FBG temperature sensors (Manuscript ID: micromachines-1039508). Those comments are all valuable and very helpful for revising and improving our paper, as well as the important guiding significance to our researches. We have studied comments carefully and have made correction which we hope meet with approval. Revised portion are marked in red or blue in the paper. The main corrections in the paper and the responds to the reviewer’s comments are as flowing:
Point 1. One of the main conclusions that the authors report is that “The smaller the heat storage capacity and the larger the thermal conductivity, the faster the response of the sensor.” This seems to partially agree with the analytical solution presented by the authors, where the copper fill presents a faster response than the other two, although one would expect that thermal grease, with a heat conductivity 2000 times that of air would respond faster. The modeling and experimental results entirely contradict the analysis and the conclusion, but no clear explanation for this is given by the authors. This is surprising and contradictory. Please explain this.
Answer: Answer: For different packaging structures, the dimensions of the inner thimble and outer thimble are the same, and the thermal physical parameters of the inner thimble and outer thimble material are exactly the same. so the difference in response speed of the sensor comes from two aspects: 1, the thermal physical properties of the filling material; 2, the heat transfer between the filling layer and the inner and outer thimble characteristic. In different environments, the heat transfer coefficients of enclosed air and grease are quite different. We consider the air flow state and the filling state of the thermal grease (the thermal grease is not in good contact with the casing), so the heat transfer coefficient of the thermal grease is estimated to be about 300 times that of air. The thermophysical parameters of the packaging material and the sensor geometry are taken into the time constant formula, and it is obtained that the time constant when filled with grease is approximately 2.5 times that of air. Therefore, the time constant of the grease medium sensor is larger than that of air.
For calculating the time constant of the copper medium sensor, the contact state between copper and stainless steel, and between copper and copper cannot be estimated, and the heat transfer coefficient cannot be converted equivalently. So the ideal model is used (assuming that the copper is in good contact with the inner and outer thimble). However, in the experiment, there are two gaps between the copper and the inner, and between the copper and the outer thimble, the interface has a larger contact thermal resistance, and the interface heat transfer coefficient becomes smaller. The simulation model is based on the experimental structure and size. So the response speed of copper is the best in theoretical calculations, while in the modeling and experimental results, the response speed of copper has deteriorated.
(line 331-334, marked in red)
Point 2. The authors pose the main question that sensor robustness through encapsulation and fast response times are contradictory. However, the experiments only focus on sensor response. What is the effect of the different studied encapsulations materials in the robustness of the sensor? It seems like this information is necessary if the sensor response is only improved by a few seconds but the robustness is compromised much more significantly. Additional experiments may not be needed but justification may be helpful.
Answer: In order to protect the fragile grating and isolate the influence of the applied stress on the wavelength drift, the fiber grating temperature sensor must be encapsulated in a protective structure. Generally, FBG temperature sensors are packaged with a single-layer metal sleeve, but the robustness and response speed of this type of sensor are contradictory.
The double-thimble temperature sensor we designed is more robust than a single-layer thimble (the outer thimble isolates stress to ensure robust characteristics; the inner thimble isolates the stress transmitted by the outer thimble and the stress of the filling layer, thereby enhancing robustness ). Moreover, in the double-thimble structure, different filling materials only affect the temperature response characteristics, and will not affect the robustness (the sensor strength is determined by the stainless steel thimble, and the inner thimble isolates the stress effects of the three filling materials).
Point 3. Briefly describe in the introduction what FBG temperature sensors are and how they work so that the readers can put this study in context and better understand the challenges presented by the authors.
Answer: In the introduction, the principle and working background of fiber Bragg grating are introduced.
(line 18-23, 28, marked in blue)
Point 4. What follows are just some grammatical fixes that could help improve the quality of the paper. This list is not exhaustive; I recommend extensive review of grammar.
Answer: We have corrected the errors pointed out by the reviewers, and checked on the grammar and spelling of other parts of the manuscript. Revised portion are marked in red in the paper.
Line 18 - FBG temperature sensors “are” instead of “were”
Replaced “were” with “are”. (line 23, marked in red)
Line 22-23 – Does the temperature sensor have excellent mechanical strength? Or is it the packaging that provides excellent mechanical strength? Please modify the sentence to clarifty.
we have re-written this sentence (line 29-30, marked in red).
Line 24 - in order to achieve response (not respond) to the ambient temperature rapidly
Replaced “respond” with “response”. (line 23, marked in red)
Line 32 – “A few researchers” or “research studies”
“A few researchers” (line 39, marked in red)
Line 34 – delicate instead of delicat
The spelling error has been corrected. (line 41 marked in red)
Line 36 – what is an RFBG sensor?
“regenerated FBG”. (line 40 marked in red)
We checked the abbreviations that appeared in the text and revised them according to the specifications.
Line 37 and throughout – Place a space between numbers and units (i.e 9 s instead of 9s)
We checked and corrected such errors
Lines 51-53 – incomplete and run-on sentence, please re-write.
we have re-written this sentence (line 59-61, marked in red).
Line 67 – period should be replaced by a comma.
we have re-written this sentence (line 79-83, marked in red).
Line 54 – Uses three types of materials
We checked and corrected such errors
Line 55 – what does “respectively” refer to at the end of the sentence?
we have re-written this sentence. (line 59-61, marked in red)
Line 64 – Did the authors fabricate the sensors? Or where they purchased commercially? This is not clear. If the focus on the paper is also on the grating structure it may be helpful to include a schematic explaining how the device works in relation to equation 1, 2 and 3.
We have added a diagram named “Transmission and reflection spectrum of FBG”, and explained the content of this part. (line 76-78, marked in red).
Line 66-69 – Please fix the punctuation of these three sentences. It is difficult to understand the meaning.
we have re-written this sentence (line 79-83, marked in red).
Line 71- “… , which ultimately affect …”
we have added the word “which” before the word “ultimately”. (line 84, marked in red).
Paragraph after Line 72 – This is said twice: “The capillary has good thermal conductivity and flexibility”
We corrected such errors (line 94).
It is not clear to me what the “reserved length” is. Please explain.
We have explained the “reserved length”. (line 101-102, marked in red).
Line 80 – In contrast, an encapsulation ….
We checked and corrected such errors
Line 86 – I fail to see the relevance of this sentence in lines 84-86. Please, help the reader make the connection and clarify how this relates to an air medium: “According to the Grashof criterion [19], fluid flow is restricted in a limited interlayer space and the development of fluid thermal boundary layer is disturbed, when Grd 2430, the heat transfer method of flfluid is heat conduction.”
we have re-written this sentence (line 114-116, marked in red).
Line 97 – Ideally, the applied stress should have no effect on the wavelength shift of the fiber, and the bonding method of the fiber should isolate the apparent thermal strain of the metal thimble. The wavelength shift should only related to the ambient temperature [20].
we have re-written this sentence (line 127-129, marked in red).
Line 103 and others – The correct abbreviation for Equations should be Eq. and not Equ.
We checked and corrected such errors
Line 108-111 – Please rewrite sentence, it is difficult to understand the meaning.
we have re-written this sentence (line 144-146, marked in red).
Line 120 – Both cases change lB , such that the…
we have re-written this sentence (line 155-156, marked in red).
Line 122 – The word because should not be capitalized. The sentence is long and difficult to understand, please break up in smaller ones.
We checked and corrected such errors(line 157, marked in red).

Round 2
Reviewer 2 Report
Thank you for addressing all my comments. I believe that authors have made all the necessary changes for publication.